# A qualitative exploratory study of UK first-time fathers' experiences, mental health and wellbeing needs during their transition to fatherhood

Sharin Baldwin,[1] Mary Malone,[2] Jane Sandall,[3] Debra Bick[4]

¹Florence Nightingale Faculty of Nursing, Midwifery and Palliative Care/ Learning and Development, King's College London/ London North West University Healthcare Trust, London, UK
²Oxford School of Nursing and Midwifery, Oxford Brookes University, Oxford, UK
³Department of Women and Children's Health, School of Life Course Science, Faculty of Life Sciences & Medicine, King's College London, London, UK
⁴Warwick Clinical Trials Unit, University of Warwick, and University Hospitals Coventry & Warwickshire, Warwick, UK

**Correspondence to**
Sharin Baldwin;
sharin.baldwin@kcl.ac.uk

## ABSTRACT

**Objectives** To develop an understanding of men's experiences of first-time fatherhood, their mental health and wellbeing needs.

**Design** A qualitative study using semi-structured interviews. Data were analysed using framework analysis.

**Setting** Two large National Health Service integrated care trusts covering four London (UK) local authority boroughs.

**Participants** First-time fathers with children under 12 months of age were included. Maximum variation sampling was used, with 21 fathers recruited. Ten of these men described their ethnic background as Indian, seven as White British, one as Spanish, one as Black African, one as Black Caribbean and one as Pakistani. Participants' ages ranged from 20 to over 60 years; completion of full-time education ranged from high school certificate to doctorate level; and annual income ranged from £15 000 to over £61 000. Non-English speaking fathers, those experiencing bereavement following neonatal death, stillbirth, pregnancy loss, sudden infant death, and fathers with existing severe mental illnesses were excluded.

**Results** Nine major categories were identified: 'preparation for fatherhood', 'rollercoaster of feelings', 'new identity', 'challenges and impact', 'changed relationship: we're in a different place', 'coping and support', 'health professionals and services: experience, provision and support', 'barriers to accessing support', and 'men's perceived needs: what fathers want'. Resident (residing with their partner and baby) and non-resident fathers in this study highlighted broadly similar needs, as did fathers for whom English was their first language and those for whom it was not. A key finding of this study relates to men's own perceived needs and how they would like to be supported during the perinatal period, contributing to the current evidence.

**Conclusions** This study provides insight into first-time fathers' experiences during their transition to fatherhood, with important implications for healthcare policy makers, service providers and professionals for how perinatal and early years services are planned and provided for both new parents.

## INTRODUCTION

The transition to parenthood can be a stressful time, with changes to lifestyles which can significantly impact on the mental health

### Strengths and limitations of this study

► Little is known about the mental health and wellbeing needs of first-time fathers. Using a qualitative methodology enabled the collection and analysis of in-depth data about the needs and experiences of first-time fathers during their transition to fatherhood.

► Use of framework analysis enabled data exploration while simultaneously maintaining an effective and transparent audit trail, enhancing the rigour of the analytical processes and credibility of the findings.

► The needs of first-time fathers reflected a range of ages, ethnic groups, education levels and income. While this provided a much broader understanding of fathers' mental health and wellbeing needs across different groups of men, it is acknowledged that this study was not representative and demographic differences in the small sample were not associated qualitatively with any specific views or experiences.

► Young fathers (under 20 years), unemployed fathers and those from lower socioeconomic groups were under-represented in the study sample.

► As this study was planned to be exploratory in nature, findings cannot be generalised to the wider population. However, findings have highlighted issues that may be of relevance for first-time fathers in other settings.

and wellbeing of both parents.[1–3] Positive mental health is defined as 'a state of wellbeing in which the individual realises his or her own abilities, can cope with the normal stresses of life, can work productively and fruitfully, and is able to make a contribution to his or her community' (WHO, pXIX).[4]. Rather than just focusing on prevention and treatment mental illness, the importance of promoting positive mental wellbeing has been highlighted by the Royal Society for Public Health in the UK.[5] Men's mental health and wellbeing during this period continues to be under-researched, with new fathers' health needs frequently unmet.[6] A systematic review

reported a prevalence rate for anxiety in men to range between 4.1% and 16.0% during their partner's pregnancy and between 2.4% and 18.0% during the 6/8-week postnatal period.[7] Reviews have found that depression affects 8%–10.4% of fathers between the first trimester of their partner's pregnancy and 1-year postpartum.[6 8] Research from Denmark[9] and the USA[10] showed that new fathers' depression rates were double the national average for men in the same age group who were not fathers. It has also been suggested that the incidence and prevalence of paternal mental health problems may be much higher than currently reported, as screening tools used to identify maternal mental health problems may not be as reliable when applied to men.[11]

Depression and anxiety in fathers during the perinatal period can affect their working and short-term memory loss,[12] and negatively impact on their ability to undertake aspects of their paid employment.[13] It can also have a profound impact on relationships with their partner and child.[14 15] Mental health problems in fathers are not only confined to the man, but are associated with cognitive, emotional, social and behavioural problems in their children.[16–19] A recent study of over 3000 families in the UK identified a link between postnatal depression in fathers and an increased risk of depression in their daughters at age 18.[20] While the available evidence suggests that the rates of mental health problems in new fathers and impacts on their families are widespread and persistent, UK policies for maternal and child health services do not currently address this.[21] To support men's mental health and wellbeing during their transition to fatherhood it is essential to better understand their experiences and the specific needs they may have during this period.

A recent qualitative systematic review undertaken by the authors identified three main factors that affected first-time fathers' mental health and wellbeing during their transition to fatherhood: the formation of the fatherhood identity, competing challenges of the new fatherhood role and negative feelings and fears relating to it.[22] In addition to these findings, the review highlighted a number of barriers and facilitators to fathers accessing timely and appropriate support, with several areas highlighted for further research. The fathers in the primary studies included in the review lacked ethnic diversity, and only included first-time resident fathers (those residing with their expectant partner, or their partner and child), with the mental health and wellbeing needs and experiences of first-time fathers from different ethnic and cultural backgrounds, and non-resident fathers, unknown. In the review the credibility of the finding relating to 'what fathers want' was rated as 'low' by the reviewers, based on the ConQual criteria,[23] suggesting caution should be applied to implementing findings into practice. The review findings suggested that *to better support first-time fathers' mental health and wellbeing during their transition to fatherhood it was important to establish what support new fathers want, and what interventions would be acceptable to them* (Baldwin et al, p2144)[22].

The need to investigate expectant and new fathers' information needs during the perinatal period has been identified by others in the field.[24] There is a clear need for research into the type of support new fathers want, how this is provided, who provides it and when would be the optimal time in the perinatal period to offer support. Another aspect that remained unclear in our systematic review was with regard to routine mental health screening for new fathers,[22] suggesting further qualitative research in this area to ascertain men's perceptions and receptiveness to mental health screening.

This qualitative exploratory study was designed to develop an insight into first-time fathers' mental health and wellbeing needs, focusing specifically on the gaps identified in the systematic review.[22] Non-resident first-time fathers (those not residing with the partner and child) and first-time fathers from different ethnic and cultural backgrounds were included to provide a broader understanding of fathers' mental health and wellbeing needs during their transition to fatherhood.

The Consolidated criteria for reporting qualitative research (COREQ) guidelines[25] informed the reporting of the study.

The main aim of the study was to consider how men experienced first-time fatherhood and what their perceived mental health and wellbeing needs were during this period. This included how men prepared for becoming a father; how it impacted on their emotional wellbeing; how they coped with the changes; what support/resources they accessed; how they perceived the support from health professionals; what were the barriers and enablers to accessing support; and when would be the best time to receive support or information about emotional wellbeing relating to becoming a father.

## METHODS
A qualitative approach was used to address the study aims and objectives. Choosing the right qualitative approach was important. A pragmatic approach was necessary, based on the research questions of interest rather than alignment with a specific epistemological stance.[26] After careful consideration, a decision was made to use qualitative approach informed by framework analysis.[26]

### Study setting
Four London administrative districts (known as boroughs in the UK) (two inner and two outer cities) whose population healthcare needs are served by two National Health Service (NHS) organisations were selected as study sites, to support the recruitment of a diverse group of fathers. Each site serves diverse socio-economic and cultural populations, with minority ethnic groups representing 44%–69% of the overall total population of the borough selected.[27]

| Table 1 | Study inclusion/exclusion criteria | |
| --- | --- | --- |
| **Inclusion criteria** | | **Exclusion criteria** |
| ► First-time fathers with children under 12 months<br>► Resident fathers (biological or non-biological)<br>► Those living within the health catchment area of the two NHS sites | | ► Non-English speaking fathers<br>► Fathers experiencing bereavement following neonatal death, stillbirth, pregnancy loss, sudden infant death<br>► Fathers with existing severe mental illnesses such as schizophrenia, schizoaffective disorder, personality disorders, major depression and bipolar disorder |

NHS, National Health Service.

## Participants

Twenty-one first-time fathers were recruited between September 2017 and February 2018. In the UK, health visitors, who are specialist community public health nurses, provide a routine contact to new families referred to as the 'new birth visit' around 10–14 days after birth. During these contacts, health visitors offered invitation letters and study leaflets to all first-time fathers who met study inclusion criteria (Table 1). In cases where the father was not present during the contact, these were offered to their partners. Local father's groups, GP (family doctor) practices, health centres, children's centres, nurseries and child health clinics were asked to display study posters and disseminate leaflets.

Only first-time fathers with children under 12 months of age were included. Maximum variation sampling was used, a method that explores important shared patterns across cases that emerge out of diverse variations and heterogeneity.[28] This sampling method ensured diversity in location, ethnicity, age, religion, education levels and social class, where possible.

Initially 25 men interested in participating in the study contacted the first author, of whom they had no prior knowledge. After discussing the inclusion/exclusion criteria, 21 fathers who all met the criteria were included. None meeting the inclusion criteria refused to take part. Participation was voluntary and written informed consent was obtained from each participant (see online supplementary appendix A). Table 1 outlines the study inclusion and exclusion criteria.

## Data collection

Face-to-face in-depth interviews were carried out by the first author until no new information was forthcoming and data saturation reached.[29] A topic guide (see online supplementary appendix B) was developed to provide structure and focus to the interviews, which were audio-recorded and transcribed using an approved transcription service. Participants were offered an opportunity to check their interview transcript for accuracy and provide feedback prior to analysis. Most interviews were carried out in the participant's home setting, one was undertaken in a health centre, one in a hotel lounge and one in a university setting. In two cases, the participant's partner was present during the interview. The duration of the interviews varied between 12 and 52 min, with the average being 28 min.

Field notes were written after each interview to record aspects of the interview that may not be captured on the recording such as environment, context, general observations and thoughts.

## Data analysis

Data were analysed by the first author (SB) using framework analysis and the five steps of data management for thematic analysis as described by Ritchie et al,[26] namely: familiarisation; constructing an initial thematic framework; indexing and sorting; reviewing data extracts; and data summary and display. The findings were discussed among the research team (all four authors) at each stage and there were several iterations of this process, before the final nine categories were developed and agreed by all authors. NVivo (V.11) was used to facilitate this process.

## Research team and reflexivity

The research team consisted of the first author (SB), who undertook all aspects of this study, with support from three members of her supervisory team (DB, JS, MMM). The risk of personal bias was acknowledged, especially being a female researcher exploring men's experiences. For this reason, it was important to have active involvement of a patient and public involvement (PPI) group in all aspects of this study.

## Patient and public involvement

A group of fathers influenced the focus of the study and research design. Contact with these fathers was made through a local fathers' group in a children's centre, which included fathers of varying ages from diverse cultural and ethnic backgrounds. Feedback from these fathers during the development of the study helped influence the research question, study design and data collection method. Following this, a PPI group of four first-time fathers was established to provide expert PPI to all aspect of this project. They were involved in the development of all research documents, the recruitment strategy, and consulted about the data analysis process, implications of study findings and dissemination approaches. The Fatherhood Institute, a leading non-profit organisation for fathers and fatherhood in the UK, acted as specialist

| Table 2 | Participant characteristics | | | | | | | | |
|---|---|---|---|---|---|---|---|---|---|
| Participants' pseudonyms | Age | Ethnicity | Religion | First language | Employment | Income | Education | Living with mother | Baby's age |
| Neil | 30–34 | Indian | Hindu | English | F/T | 61K+ | MSc or PhD | Y | 7 months |
| Dev | 30–34 | Indian | Hindu | English | F/T | 15–30K | Degree | Y | 8 weeks |
| Arjun | 30–34 | Indian | Hindu | English | F/T | 46–60K | Degree | Y | 6 weeks |
| Raj | 35–39 | Indian | Hindu | Gujarati | F/T | 15–30K | Degree | Y | 12 weeks |
| Jay | 35–39 | Indian | Hindu | Hindi | F/T | 46–60K | MSc or PhD | Y | 7 weeks |
| Miguel | 40–44 | Spanish | Christian | Spanish | F/T | 15–30K | Degree | Y | 4 weeks |
| Ravi | 35–39 | Indian | Hindu | English | F/T | 46–60K | Degree | Y | 8 weeks |
| Krish | 40–44 | Indian | Hindu | English | F/T | 46–60K | Degree | Y | 3 weeks |
| Tom | 35–39 | White British | Christian | English | F/T | 31–45K | Degree | Y | 8 weeks |
| Ahmed | 35–39 | Pakistani | Muslim | English | F/T | 46–60K | MSc or PhD | Y | 3 weeks |
| Lee | 35–39 | White British | No religion | English | F/T | 31–45K | Degree | Y | 9 months |
| Lloyd | 20–24 | Black Caribbean | Christian | English | F/T | 15–20K | GCSE | N | 6 months |
| Simon | 30–34 | White British | Christian | English | P/T | 15–30K | Degree | Y | 4 weeks |
| Charlie | 30–34 | White British | No religion | English | F/T | 31–45K | MSc or PhD | Y | 4 months |
| Sanjay | 35–39 | Indian | Hindu | English | F/T | Not revealed | Degree | Y | 10 weeks |
| Adrian | 30–34 | Black African | No religion | English | P/T | 15–30K | MSc or PhD | Y | 7 weeks |
| Sam | 30–34 | White British | No religion | English | F/T | 61K+ | Degree | Y | 9 weeks |
| Akash | 30–34 | Indian | Sikh | English | F/T | Not revealed | MSc or PhD | Y | 6 weeks |
| Richard | Over 60 | White British | No religion | English | F/T | 15–30K | A levels | N | 8 weeks |
| David | 35–39 | White British | No religion | English | F/T | 61K+ | MSc or PhD | Y | 6 weeks |
| Ali | 30–34 | Indian | Muslim | Tamil | F/T | 61K+ | MSc or PhD | Y | 8 weeks |

advisors to the project and will be involved in the dissemination process.

### Ethical considerations

The study was conducted in compliance with the Research Governance Framework for Health and Social Care and Good Clinical Practice. All interviews were carried out on a voluntary basis and participants could withdraw from the study at any stage, although none chose to do so. The interviews were transcribed with the principle of anonymity in mind and a confidentiality agreement was in place for the approved transcribing service used. Pseudonyms have been given to all participants in the illustrations used, to protect their identity (see table 2).

### RESULTS

### Sample characteristics

Participants' ages ranged from 20 to over 60 years; completion of full-time education ranged from high school certificate to doctorate level; and annual income ranged from £15 000 to over £61 000. Two men did not reveal their annual income, and two worked part-time while the rest were in full-time paid employment. Ten men described their ethnic background as Indian, seven as White British, one as Spanish, one as Black African, one as Black Caribbean and one as Pakistani. For four

men, English was not their first language (but translators were not required). The interviews were carried out at various points during the first year after their child's birth, ranging from 3 weeks to 9 months. Two fathers were not residing with their partner and baby at the time of the interviews. See table 2 for full participant characteristics.

Nine major categories pertaining to fathers' experiences and perceived mental health and wellbeing needs were identified from the data:
1. Preparation for fatherhood.
2. Rollercoaster of feelings.
3. New identity.
4. Challenges and impact.
5. Changed relationship: 'We're in a different place'.
6. Coping and support.
7. Health professionals and services: experience, provision and support.
8. Barriers to accessing support.
9. Men's perceived needs: what fathers want.

### Preparation for fatherhood

When it came to antenatal preparation for the birth, some men described that they had not prepared at all, while others had used various approaches. One father talked about focusing on the 'practical' aspects of infant care:

I made sure I had clothes, Moses basket, things like that, all in place so it's not a rush [Lloyd].

While another father reported to be 'over prepared' as he had sought to gather so much information:

We did antenatal classes, we did hypnobirthing classes, we did a lot of reading, we had a good group for our NCT classes, so we used to message each other, and that sort of thing [Akash].

The experiences of fathers who attended antenatal classes varied. Some sessions only focused on 'normal' births, while others the practical aspects of new parenthood and none addressed the 'social, mental aspects of it' [Neil]. Those who felt included by the health professionals found the sessions helpful and prepared them well for what to expect from the birthing process. Prospective fathers also identified the opportunity to meet 'like-minded parents' as a major attribute of antenatal classes [Tom].

Some men however did not have the opportunity to prepare, either lacking time to read information offered to them or being unable to attend antenatal classes as times were inconvenient, or because they were not invited or unaware of when sessions (which may have supported their preparation) were taking place. Lack of flexibility in the workplace was a barrier for some men to attending antenatal appointments or classes.

### Rollercoaster of feelings

Men described a range of feelings relating to becoming a first-time father, with emotions ranging from happiness and excitement to apprehension and stress in the antenatal period, with the baby not feeling 'real' during their partner's pregnancy for many.

#### Mixed feelings

One father described his emotions about fatherhood as:

a rollercoaster ride…we've got a long way to go yet until the baby arrives in this world and having that mixed emotions, really, so there's been stressful times, ………but there's also been times where we've been looking forward to it. [Krish]

Feelings of excitement along with apprehension about being a good father were a common theme:

Excitement was probably the first thing that I felt … it was a little bit of, kind of, apprehension, as in how - what will I need to, kind of, do in terms of being a dad, and will I be able to, kind of, cut the mustard, in terms of being a dad, and that type of thing.[Neil]

Feelings of apprehension and nervousness appeared to be related to the 'unknown' about becoming a father and worries about their partner and baby's health and well-being, which one man described as being 'pretty scary, overwhelming, life-changing' [Lee].

To describe the positive emotions they experienced about impending fatherhood, men used phrases such

as 'over the moon' [Krish], 'rewarding' [Sam], 'proud' [Richard], 'happiness coming from inside' [Raj], 'awesome feeling' [Jay], 'feel absolutely complete' [Miguel], 'brimming with love and joy' [Lee].

#### Not real

For many men their baby did not seem 'real' during their partner's pregnancy as they could not see the baby or physically feel what their partners were feeling. Some men described not feeling like a father until after the birth of their baby:

Even though the baby was there, you can see the bump, you can see, you know, the baby moving around inside, to me, it wasn't there. Yeah, it wasn't real. It's only until she was born… [Dev].

Another father stated that 'it was something that I couldn't quite process until it [the birth] actually happened' [Charlie].

### New identity
#### Sense of accomplishment and personal growth

Men described their experience of becoming a father as a sense of accomplishment and personal growth. Fatherhood was a positive change which made them feel more secure and confident.

I think, for me, at the moment, it has been a very, very positive change…this thing is going to help me to be a better person, a better father and, yeah, it's good for me. I feel more secure. I feel as well as that more confident. I was fine before, you know, but now I feel like - it's I feel complete [Miguel].

Being able to father a child and start a family was also an important aspect of this change:

… it is a good feeling in a way, 'cause I thought I was - I'm not a confirmed bachelor, certainly a confirmed fatherless person, but a person who wouldn't have children [Ricard].

Several participants felt that becoming a father made them 'stronger' and more 'resilient' as they had to learn how to cope with the demands of early fatherhood by themselves.

You really had to just get on with it, and I guess that makes you stronger as a person for yourself [Ravi].

#### Changed person

The new fatherhood identity meant a changed lifestyle, which came with different responsibilities, changed priorities and an altered mindset for many of the men interviewed. New fathers described that they now prioritised their partner and baby's needs over their own, as one father said:

your own needs really go out of the window [Ravi].

While another said:

I will compromise all the things for my baby and my family, to be honest….I'm not worried about my things. I'm worried about all my baby's and my wife [Raj].

The new fathers wanted to spend more time with their family rather than going out socially, and acknowledged additional responsibilities they had to take on, to look after their partner and new baby:

You'd rather be at home with the baby. Save a bit of money as well for the baby…in terms of your mind set changes a bit, as well…so you start thinking differently. Now you've got boundaries, yeah? You can't cross them boundaries. [Dev]

### Challenges and impact
When asked, all fathers described the impact of new fatherhood on their health and wellbeing in terms of positive and negative aspects.

#### Challenges relating to labour and birth
Some of the men interviewed found the experience of labour and birth quite stressful, mainly due to not knowing what to expect, as Sam explained:

I just had no idea how long labour could be, 'cause R went into labour and had a basically, a three-and-a-half-day labour. You know, in films it's, like, half an hour labour and, you know, I consider myself reasonably well educated and I didn't really know the detail [Sam].

#### Tiredness, exhaustion and stress in early fatherhood
A lack of sleep, missing meals and having to balance work commitments with family life were commonly reported triggers for tiredness and stress:

It's tough 'cause you've got - you're not sleeping, you're missing meals and like, I think those - that, for me, just missing the sleep and missing the meals, makes me more cranky and you just become a bit more snappier [Arjun].

Many new fathers found it very difficult to balance work and home life. This stress was often exacerbated for those who wanted to remain at home with their new family rather than be at work. Some new fathers described concerns that they were 'missing out' on their infant's early life because of going to work:

I'm just more concerned about just missing out, as well as just being able to be there, close at hand is, sort of, my concern [Adrian].

The repetitiveness of the cycle of caring for their baby and being in paid employment meant that new fathers had little time for themselves, as one father described:

you give her a feed and you put her to bed and then you unwind, if you can or you don't, and then you go

to sleep. And then you'll know like at 12 o'clock or 3 o'clock she'll wake up and you'll have to feed her. And that's the really difficult time. … 'cause you're exhausted from work, and then like, during that period you know something's going to happen. So, you have to care for her then and then, you have to wake up again at 6 o'clock to get ready for work again. And then, you're doing your eight or nine hours at work and you come back and it's - you're doing that same cycle. [Arjun]

#### Increased worries and pressure
Men felt responsible for taking good care of their partner, for example taking over some of the household chores that their partner would normally do. Some men described the impacts of the additional financial strain of having a baby, but wanted to adequately fulfil their role as a provider:

…there's definitely more pressure on the man, because there's - the second income has just disappeared from the household and all the rest of it. And there's more pressure to get things done and make sure that you're providing for them… [Sanjay].

Worries relating to the health and wellbeing of their baby and partner were apparent in many men's accounts. They worried about 'knowing the right thing' and 'getting it right' as a father. Ahmed described how this impacted on him:

Worry all the time, just that the baby's okay, that, you know, is your wife getting everything she needs and then, is the baby therefore getting everything she needs? Is she going to be healthy? [Ahmed]

#### Emotional impact
The additional stress resulting from the tiredness and pressure to provide for their family impacted negatively on several fathers. One man described how it made him 'more cranky' and 'a bit more snappier' [Arjun], while another talked about how difficult and frustrating it was not having the right information and knowing what to do in the early weeks following birth, which resulted in negative emotional impacts:

…it can bring you down very, very fast. Very difficult situation sometimes and yeah, an element of you can go into some form of a depressive state where, you know, you start to get frustrated at each other, because you're both unaware what to do and your children are crying and it's like, what do we do? [Ravi]

When not able to give their partner a break by comforting their baby, fathers talked about feeling 'useless' [Sam] and how 'demoralising and demotivating' [Neil] the experience was. Some fathers also expected an instant bond with their baby, and when this did not happen they found the experience quite challenging:

.... particularly in the first week when the baby doesn't recognise you, of just not feeling like they - you can make them feel better. I would say that's probably a challenge [Sam].

Sam also described as feeling 'useless' and as 'bit of a spare part' when not able to stop his child from crying.

Other fathers described not knowing how to help or support their partner who was breastfeeding as being challenging, a situation which became worse when the woman experienced breastfeeding difficulties or the breastfeeding 'did not go to plan':

Well, I think the most difficult thing that we faced was breastfeeding, and there was a lot of information that was given and it was all, kind of, geared towards how breastfeeding is great for your child, and all of those kind of things, but it was none of the, kind of, practical tips of what to do once things start going wrong, in the sense that your child may not know how to latch. So, as a dad, what can you do to, kind of, support that? [Neil]

### Rewards of fatherhood

Although men described new fatherhood as challenging, they also described it as being fun, enjoyable and rewarding. They particularly enjoyed being able to interact with their child and watching them grow:

It's been brilliant. I've never known anything as joyfully rewarding in my whole life, it's brilliant. Having a baby, the giggles when you do something silly and yeah, he's just great, brilliant. [Lee]

### Changed relationship: we're in a different place

Other positive aspects to new parenthood included men describing their relationship with their partner as stronger since the birth of their baby:

it's definitely brought us closer together. Made us, you know, more in love, stronger as a couple......... We're being very supportive of each other and that has helped build a, you know, helped strengthen our relationship [Simon].

Some men talked about other changes they noticed in their relationship with their partner. These included not having time to spend with their partner, arguing more, being less intimate sexually and their partner being more irritable:

I probably argue a bit more and that's probably just due to my tiredness. [Arjun].

What possibly has suffered is that in some way, sexually, we haven't been as intimate [Richard]

However, most men recognised that these changes were due to tiredness or the demands of new parenthood and that this was just a different phase of their relationship:

we've just moved on. We're in a different place ...... Our relationship is different now, but we still have fun and we get on most of the time. I mean, we argue once or twice a week maybe, but yeah, generally, we're getting on with each other. [Lee]

### Coping and support

Men described a variety of internal and external resources which enabled them to cope with the 'challenges and the impact' of new fatherhood discussed earlier. The need to cope alone was a common theme. One father said:

I tend to keep it in myself so, you know, I battle it myself, in terms of being - so, you know, lack of sleep, you know, that - my first week back at work and I'm there falling asleep on my desk. But yeah, I don't show it, I just, kind of - oh, he's crying - but I just, kind of, battle in continually......I won't share my, kind of, worries and thoughts. I tend to fight it inside me and think, okay, you know, okay, I'm - you know, I've got this, what - you know, whilst, you know, keep it in my head ...I won't show it to, you know, my wife ...I won't show her that I'm feeling that way. I just, kind of, put a smile face on, but then tackle it behind the scenes. [Krish]

Another father talked about not wanting to burden his partner with his own worries because she was 'going through much more' than him and that she would not be able to relate to his concerns [Adrian]. He also felt it was socially unacceptable for men to talk about their difficulties and therefore he just had to 'get on with it'. For many men it was just a matter of 'getting on with it' and learning through 'trial and error'.

Men sought information about the practical aspects of infant care from several online resources, including NHS websites, Netmums, Mumsnet and Baby Centre. Some were however cautious about trusting online sources unless they were from NHS websites [David] and found it frustrating not knowing which of 'Google's thousand solutions would be relevant to their own situation' [Ravi].

While some men did not share their own difficulties relating to the challenges of new fatherhood with their partners, others felt comfortable to do so, and worked together with their partner as a team to manage the demands of early parenthood. They shared daily tasks and took it in turns to care for the baby.

External family support was evident in a number of fathers' accounts, which included support from their own parents, siblings and members of their extended families. For some however, this additional support was not available due to their family living in another country or that the advice they received was not relevant to their own situation. Some men sought support from friends but described the conversations as casual and light-hearted. As Lee explained: 'we blokes are rubbish at talking'.

Friends who were not parents themselves were even less likely to relate to the new father's situation. Similarly,

connections made with other fathers through parenting support groups, such as the National Childbirth Trust (NCT), were reported to be helpful, but for many the conversations were still 'light-hearted' and 'not necessarily very deep and meaningful' [Charlie].

## Health professionals and health services: experience, provision and support
### Experience
Men's experiences of contacts with the relevant health professionals and health services varied. Some men described their interaction with midwives and health visitors as 'really good' [Lloyd], 'always been helpful' [Arjun] and felt they 'were in safe hands' [Lee]. Their experiences were mainly positive when health professionals enquired about their own wellbeing and made them feel included, as one father described:

> the midwives, yeah, made me feel really, kind of, you know, quite an important - as a, sort of, birthing partner, quite a, sort of, important piece of the jigsaw, really [Tom].

However not all men had such experiences, as another father explained:

> …everybody I've been in contact with has, … sort of, been in the mindset of treating you like you're a bit of a tool… I was putting the poppers on and obviously [while dressing the baby], there's loads of poppers and because I was a dad putting the poppers on in a room full of mothers, they're [health professional] like ha, ha, you know, look, dad's struggling with - you know, that's - it's general humour that people are quite comfortable with [Adrian].

Despite feelings of exclusion, many fathers simply accepted this as the status quo. They validated this with reasons such as their partner's needs being greater than theirs, and health professionals not having enough time due to their workloads:

> they were concentrating on M and the baby. No, I don't think - did they ever ask me? I don't think so, no. They're busy, aren't they? [Lee].

One father [Tom] felt that the health visitor was mainly there to check up on them from a child protection point of view and was unable to answer the questions they had asked. This made him question the knowledge base of the health professional. Others felt they received conflicting advice from different health professionals [Akash] and some referred to a lack of coordination between members of the perinatal health services [David].

### Provision and support
A lack of adequate facilities for fathers in hospital labour wards was raised. There was no provision for fathers to sleep, eat or wash, even when they had to stay with their partner overnight:

> during the hospital stay, you know, there's, sort of, no provision made for partners…I mean, you're allowed to sleep on the floor, which is lovely, and you know, doable, but … there are no facilities where one can take a shower, for example, you just have to go home [Simon].

This was particularly problematic when they had to stay there for long periods, supporting their partner in labour. Men also found it difficult when they were asked to leave overnight following their baby's birth as Ali explained:

> …when the baby is born they don't allow you to stay in the hospital, so maybe there can be some improvements over there [Ali].

When it came to accessing support for their own mental health and wellbeing, most fathers said they would only approach health professionals as their last port of call and the GP (family doctor) would be their professional of choice, as David explained:

> I'd consider seeing a GP for a referral, but I wouldn't approach the Maternity Services for that stuff. I wouldn't ask the Health Visitors or other people we see at the GP.

## Barriers to accessing support
The men interviewed described a general lack of appropriate support and information for new fathers. Only being able to obtain information through attending antenatal classes, which were more tailored to women, was insufficient. Some struggled to get appointments with their GP for their own mental health needs in a timely manner. Many did not know if there was any specific mental health support available to fathers. As one interviewee stated:

> You don't really know it's accessible to you [Arjun], while another said: I don't know where you'd actually go for that kind of support, necessarily. [Sanjay]. Furthermore, ambiguity about how to access mental health support in the NHS was described [Sam].

Some fathers questioned the training of GPs in dealing with fathers' mental health issues [David], or that GPs were too 'stretched' to deal with such issues [Sam]. Men feared taking up health professionals' time with their own mental health worries and avoided seeking help: 'I feel like you really are aware - with that in mind, you really are aware that you're taking up somebody else's time if you are to be in that position, and it's like, you know, I don't want to bore you with my troubles' [Adrian].

Most men were not asked about their mental health and wellbeing by health professionals at any contacts in the perinatal period. They viewed the health professionals as being mainly there for their partner and not for them. Fathers' opinions were perceived by those interviewed as not important. Akash described that involving the father would be a 'foreign concept':

…no one really asks you how the father is doing, it's all about the baby and the mum. So, yeah, it's just a foreign concept, I think. [Akash]

It was apparent that there were several perceived barriers to men accessing support for their own mental health and wellbeing, including a general view expressed by the interviewees that it was difficult for men to talk about mental health problems. As perinatal services were perceived as being mainly for women, men often felt uncomfortable in female dominated groups such as those for breastfeeding support, postnatal support and mother and baby groups. Some men felt that it was culturally and socially unacceptable to talk negatively about fatherhood experiences or to admit to experiencing difficulties: '…feel a bit ridiculous if you're saying, 'Oh no, I'm finding it really difficult" [Sam].

There was also a general fear of being perceived negatively by work colleagues, friends and family if a mental health problem was identified, as men referred to the stigma of mental health:

I guess, it's that fear of worrying about well, if you went and then seek help, how would your company see that? How would your friends and family see that? Is that something you want to disclose? … I think that sometimes can be the making or breaking point for someone where, if you do need to seek the advice, but you don't because of other fears, it then means that you're learning to cope with it in different ways [Ravi].

### Men's perceived needs: what fathers want
#### Better preparation for fatherhood
The men interviewed wanted to be better educated on what to expect during their partner's labour, and their role as a birth partner. They described needing more information on the physical and emotional demands of parenthood in the early days and weeks after birth:

if, you know, you had someone who said I know you would've heard this before, but there will be a serious lack of sleep, to the extent that you will feel quite disorientated. You may have times where you struggle to bond with the child. You just need to be aware that that is very normal. [Sam].

Some felt it would be helpful to have antenatal classes with a specific focus on dads, either delivered together for couples or for fathers only [Ahmed].

#### Better access to information and services
Men wanted access to correct, relevant and up-to-date information on the practical aspects of infant care, the challenges relating to new parenthood and to know what support services were available specifically for dads. Although men accepted being excluded by health professionals, they felt that they should be asked about their own mental health and wellbeing and offered the same level of support as women. Most said they would be willing to speak to health professionals about their mental health and considered that it would be better to be given an

opportunity to seek help, rather than face having to deal with it themselves:

100%, I would, yeah. I'd rather talk about it than bottle it up … I'd rather have the help from the get-go, than trying to figure it out myself and then stress myself out about it [Lloyd].

New fathers wanted information about 'signs' and triggers of mental health problems so that they knew 'trigger points at which you may want to, kind of, talk to somebody about it to, kind of, relieve some of your stress' [Neil].

This was particularly important for some men who felt that rather than disclose their own difficulties in front of their partner, they would prefer to have the information at hand and able to contact a health professional independently.

However, it was apparent that the men interviewed would only feel able to disclose any mental health difficulties if they knew that the health professional contact and/or assessment would include their needs and not just those of their partner and baby. Adrian described:

if I'd known that … was the focus of their visit or what have you, then maybe, yeah. But I feel like, it's - you feel it's light conversation most of the time, so you're like, 'Oh yeah, yeah, I'm fine', you know? [Adrian].

Being given an opportunity to ask health professionals questions was important as another father explained:

if the midwife comes, …even if it's a five min slot, just to have a catch-up with the dad, just to see, do you need anything? Are you doing this? …One maybe even if it's a minimum, they feel included or two that there's an opportunity to ask questions, that they might be nervous about. [Ahmed]

A number of factors were identified that fathers interviewed considered could facilitate better access to support for new dads' mental health and wellbeing. They included: knowing where to go for help, more joined up health services, father-focused information and leaflets, hospital provision for fathers on labour wards (for eating, sleeping, washing), more emphasis and priority placed on men's health by health professionals, antenatal classes to include issues fathers may face in the postnatal period, fathers to be asked questions about their own wellbeing, regular checks for fathers during the perinatal period (by health professionals) and weekend or evening antenatal appointments/parentcraft sessions for working fathers. Having the flexibility in the workplace to attend antenatal appointments was identified as a facilitator for fathers to feel more involved.

### A variety of sources of support throughout the perinatal period
When asked about sources of support, the men interviewed wanted support for their mental health and wellbeing, as well as practical aspects of caring for their partner and baby to be available in a variety of formats. This included face-to-face contacts with health

professionals, leaflets, online resources, support groups, apps and telephone advice. Support offered face-to-face or by telephone contact was preferred and seen as ideal, as Charlie explained: '… the more, sort of, personalised the contact, probably the better'. Written information needed to be brief, concise and to the point. Men also saw the value of fathers' support groups where they could learn and feel supported by more experienced fathers.

Most of those interviewed considered that this support should be available to fathers throughout the perinatal period, not just during their partner's pregnancy:

> Well, I think it during all the process, because it's important to know what is the next step you have to take and yeah, once you have made that next step, you need to know [laughs] what is the next one? So, this is a continual - continuous process [Miguel].

## DISCUSSION

This study explored the needs and experiences of a diverse group of first-time fathers, during their transition to fatherhood. It is unique in that it explored men's own perceived needs and how they would like to be supported during and beyond their partner's pregnancy. Findings highlighted the changes and challenges new fathers experienced and impacts on their mental health and wellbeing, some of which were similar to previous studies as discussed below. New findings with respect to the level of support first-time fathers wanted from health professionals and the timing of this add to the evidence, with important implications for how perinatal and early year's services should be designed and provided for new parents.

This study identified that new fathers gave a significant amount of thought both before and after their baby's birth to 'fatherhood' and what it meant for them. The fathers interviewed thought about their own and their partner's mental and physical health, and their role and responsibility as a father. They were willing to approach and use health services but were unsure if this was appropriate or if the health professionals they consulted would have the relevant skills and knowledge base to deal with fathers' mental health. That new fathers feel excluded by health professionals and unable to access appropriate information has been previously reported.[24 30–35] Questioning the knowledge-base of family doctors and health visitors in relation to fathers' mental health reported in this study was highlighted by Rowe *et al*, in an Australian study of 22 women and 16 men, where fathers suspected that the primary care health professionals were 'not qualified to emotionally help you' because their training prepared them to treat physical not mental illnesses (Rowe et al, p50, 51)[36]. This suggests a need for health professionals providing care during and after pregnancy to inform new parents about their clinical role in supporting mental and physical health, so that both parents know what to expect from them. Another gap in services related to the lack of adequate facilities for fathers in hospital labour wards, which was previously reported by Symon *et al*[37] in a survey of the experience of maternity environment and care involving 515 couples across nine maternity units in England. Further work is needed which specifically considers how fathers are treated on labour and postnatal wards, with an aim to improving resources and space for expectant and new fathers so that they, in turn, feel better able to offer appropriate emotional and practical support to their partners.

When it came to accessing support for the father's own mental health and wellbeing, most of the men interviewed would only approach health professionals as their last 'port of call'. Family doctors (GPs) would be their health professional of choice to approach, although this was likely to reflect that the fathers interviewed were unaware which other health professional was the most appropriate to seek help or advice from. Although the fathers interviewed 'accepted' being excluded by health professionals, they wanted to be asked questions regarding their own mental health and wellbeing, and to be offered the same level of support as women. Most fathers would be willing to speak to health professionals about their mental health if they knew that the service was available for both parents.

Evidence from the current and earlier studies highlights that if relevant health professionals fail to engage with fathers throughout the perinatal period, or only occasionally casually enquire about the father's health and wellbeing, they will not identify the father's mental health needs. However, if they make a point of specifying that they are there to support and consider a new father's mental health and wellbeing as well as the woman's, they may be more likely to get a response, as a father may feel more comfortable to talk about his feelings. This has major implications for the planning and content of health professional's contacts during the perinatal period which need to be more family/parent focused, rather than focused primarily on the woman and infant. Health professionals' apparent limited experience of prioritising fathers' health needs, and inability to assess fathers' mental health and wellbeing, means that men who are experiencing mental health problems are unlikely to be identified, as described in previous studies.[38 39] Both these studies suggested that fathers should be routinely assessed for postnatal depression and that health professionals need to be adequately skilled to working with and identifying depressive symptoms in fathers.[38 39]

Similarly, some studies have highlighted training needs for health visitors with regard to working with fathers to better support their mental health and wellbeing.[40 41] In a study of two focus groups (each comprising six health visitors), Whitelock[41] identified a lack of training around fathers' mental health; a lack of confidence; fears of own safety; and a lack of policies to screen fathers' mental health as barriers that prevented health visitors from assessing new fathers' mental health and wellbeing. Similarly, Oldfield and Carr[40] in a qualitative interpretative phenomenological

analysis study of three student health visitors reported that paternal mental health was not addressed in their training, resulting in the students feeling inadequately prepared to support fathers in practice. To provide the level of support identified by the fathers interviewed in this study, health professionals would need in-depth understanding of fathers' mental health and wellbeing needs, and confidence and skills to assess and support their mental health. This suggests that there is a need to ensure fathers' mental health is incorporated in student training and made available as post-registration training for all professionals involved in working with parents during the perinatal period.

Few studies to date have explored what type of support fathers would like during the perinatal period. Fathers in this study reported various levels of support relating to their preparation for fatherhood, and wanted better preparation for the birth and parenthood, similar to that of the findings of our recent systematic review.[22] They particularly wanted 'frank discussions' about the difficulties they may face in early parenthood (such as sleepless nights, exhaustion, relationship changes) and what to do when 'things go wrong'; for example when men were not able to support their partner through breastfeeding difficulties or when they did not feel an instant bond with their baby after birth, they described feeling 'useless' and as 'bit of a spare part', feelings which have also been previously reported.[42 43] Similarly, during their partner's pregnancy, some men experienced increased stress due to negative feelings about the pregnancy, the upcoming birth and the first weeks of fatherhood, consistent with other studies.[44 45] As high anxiety or depressive symptoms during pregnancy are the most significant predictors of depression in men in the postnatal period,[46] the need for better information and support for expectant fathers in the antenatal and early postnatal period is crucial. The findings of this study suggest that providing fathers with adequate information in the antenatal period to help them better prepare for changes and challenges ahead is likely to make their transition to fatherhood much more positive and may reduce their stress levels. While some literature suggests that men may show a preference for father-only groups, this is unlikely to be sustainable in the UK NHS, due to cost and staffing resources.[47] Findings from a survey of 69 expectant fathers from five different ethnic minority groups in London reported that the majority of fathers favoured joint antenatal sessions with their partner over male-only classes, and there was no preference for a male facilitator.[48] Recommendations from a large literature review by the Movember Foundation suggest that staff characteristics, skills and qualities such as being non-judgemental, male positive and empathic to men's needs are far more important than the sex of the staff.[49] Therefore, in order to work successfully with fathers, practitioners have to consider addressing fathers needs as men, as well as fathers (similar to the way in which a family-focused approach is used with women) and not just as child carers.[50]

The first-time fathers interviewed, not unexpectedly, described tiredness and exhaustion following the birth of their child, and struggles relating to balancing conflicting demands, such as spending time with their child and having to go to work, which confirms previously reported findings.[2 22 30 42 51 52] Fathers in this study described a sense of accomplishment and growth relating to new fatherhood, and often feeling conflicted between wanting to 'be there' for their partner and adequately fulfil the financial provider role as a father. The concept of 'good fathering' and fulfilling their role as 'men' has been linked to men's ability to financially provide for their family, forming part of their identity and self-worth.[22] The impact of the conflicting aspects of 'fatherhood' and internal struggles new fathers experience are important considerations to understand first-time fathers' mental health and wellbeing.

Relationship changes with their partner were of concern for some of the men interviewed. However, these changes were seen as temporary or indicated a shift or 'change' in their relationship as the couple became parents. A positive finding was that the majority of those interviewed described their relationship with their partner as stronger since the birth of their baby. This contrasts with the systematic review findings where a deterioration in couples' relationships following the birth was reported[22]; however, this may reflect different populations studied. For example, this study only recruited 'well fathers' (with no history of severe mental illness), all of whom were in couple relationships at the time of interview and possible that while some men talked about 'arguing more' after the birth of the baby, most coped well due to their strong relationship as a couple. It was apparent that those who described their relationship as becoming stronger were working together to fulfil the duties of early parenthood. According to Houts et al[53] relationship deterioration is highest among couples 3 months after birth as they are more likely to use destructive (negative escalation, use of threats and coercion) problem solving methods rather than constructive (remain engaged, issue focused and concentrate on negotiation) problem solving methods. In this study the majority of fathers were interviewed prior to 3 months following the birth of the baby, which could further explain positive findings on reported couple functioning.

A number of barriers were identified to first-time fathers accessing support during the perinatal period, including gaps in service, not being informed about the services available for fathers, being excluded by health professionals, inflexible working practices and self-imposed barriers. Father also wanted information about 'signs' and triggers of mental health problems so that they knew when to access help. These findings suggest that new fathers need to be provided with adequate information about perinatal mental health, what services are available, how to access them and which professionals they could approach for additional support.

This study identified a number of facilitators for better access to support for new fathers. New fathers wanted

support to be available in a variety of formats, with face-to-face or telephone contacts being ideal. Most men stated that support should be available to fathers throughout the perinatal period, especially from the third trimester of the pregnancy until the first few months after birth. Fathers' mental health and wellbeing needs therefore need to be considered by health professionals who provide care to women throughout the perinatal period.

Study limitations included that only volunteer first-time fathers participated, which may have resulted in recruiting those who were specifically interested in the topic area; with new fathers from lower socioeconomic groups, unemployed and younger fathers (under 20 years) under-represented. It is recognised that fathers from these groups may have variable experiences during the perinatal period, which were captured in this study. Younger fathers (under 20 years) however only make up 1.1% of all fathers in England and Wales,[54] which may explain under-representation of this group in the study.

Although all fathers included in the study had a child less than 12 months of age, it is acknowledged that the first 12 months could encompass variable postpartum experiences. Most interviews (17 out of 21) were conducted in the first 3 months following birth. Of the remaining four, two took place between 4 and 6 months postpartum and two between 7 and 9 months. Findings however provide an insight into how first-time fathers could be better supported during their transition to fatherhood, and why this is important.

There were no substantive patterns identified between the different groups of fathers. For example, there were no obvious linkages noted for ethnicity, age, income or education of participants. Resident and non-resident fathers interviewed highlighted broadly similar needs, as did fathers for whom English was a first language and those for whom it was not. As this study only included first-time biological fathers, the mental health and well-being needs and experiences of subsequent fathers, and non-biological fathers remain unknown, areas which need to be addressed in future research.

Study findings have important implications for policy makers. Men's perinatal mental health is currently not accorded the same priority as maternal mental health. Although this has been overlooked until recently, in what is being called a 'landmark move', NHS England announced as part of their 'Long Term Plan', 'Fathers/partners of women accessing specialist perinatal mental health services and maternity outreach clinics will be offered evidence-based assessment for their mental health and signposted to support as required' (NHS, p 49) [55].

While this is a move in the right direction, this study suggests *all* fathers need routine assessment and support for their mental health and wellbeing during the perinatal period, not just those whose partners are unwell.

**Acknowledgements** A big thanks to all the fathers who took part in this study, for sharing their views and experiences and also to the father advisers (patient and public involvement group) who supported and informed the study throughout. We would like to thank the two NHS trusts that participated in the study. We would also like to acknowledge the support of The Fatherhood Institute for their advisory role and the funding from the National Institute of Health Research which made this study possible.

**Contributors** SB developed and designed the research proposal, negotiated access to the study site, obtained the required approvals, recruited participants, conducted the interviews, undertook the data analysis and wrote the first draft of this paper. DB, JS and MM supervised the study, reviewed and agreed the coding framework, commented on the draft manuscripts and agreed upon the final version of the paper.

**Funding** This study was funded by a National Institute for Health Research Clinical Doctoral Fellowship (ICACDRF-2015-01-031).

**Disclaimer** SB is funded by a National Institute for Health Research Clinical Doctoral Fellowship (ICACDRF-2015-01-031). This paper presents independent research funded by NIHR. JS is a NIHR senior investigator, and JS and DB are supported by NIHR Collaboration for Leadership in Applied Health Research and Care South London. The views expressed are those of the authors and not necessarily those of the National Health Service, the NIHR or the Department of Health and Social Care.

**Competing interests** None declared.

**Patient consent for publication** Not required.

**Ethics approval** Approval was obtained from the Health Research Authority and given favourable opinion by London—Fulham Research Ethics Committee (IRAS no: 203629).

**Provenance and peer review** Not commissioned; externally peer reviewed.

**Data availability statement** Data are available upon reasonable request. All data relevant to the study are included in the article or uploaded as supplementary information.

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
