## [Reviewer comments · BMJ Open]

ARTICLE DETAILS

TITLE (PROVISIONAL)	A qualitative exploratory study of UK first-time fathers' experiences, mental health and wellbeing needs during their transition to fatherhood
AUTHORS	Baldwin, Sharin; Malone, Mary; Sandall, Jane; Bick, Debra

VERSION 1 – REVIEW

REVIEWER	Peter Gray University of Nevada, Las Vegas
REVIEW RETURNED	17-May-2019

GENERAL COMMENTS	This is an excellent qualitative study of a sample of 21 first-time fathers in the UK. The methods are described well and appropriate to the study aims. The results are organized into 9 key themes, with illustrative quotes taken from interviews provided for these themes. The work is situated well within the literature, both in the Introduction and Discussion sections. I share some additional considerations related to broader interpretation and limitations, but feel that most recommendations (e.g., to cite specific works) amount to somewhat arbitrary additions given the high quality methods and achieved aims in the present manuscript. With respect to limitations and methods, this is obviously a non-representative sample, which raises questions about generalizability, but the authors already acknowledge that. Still, it appeared that only one father was younger than 30 years of age; in the Discussion, the authors might note that some literature highlights paternal age (e.g., teenage fathers vs. older fathers in say 50s+) as a salient variable to paternal experience itself. More information about participation rates (e.g., number of invited participants, refusals, etc.) could be provided. The time-frame (first-time fathers with a child less than 12 months of age) could encompass quite variable post-partum experiences, durations since birth, etc., which is another limitation. In the Discussion, a few more contextual references could be provided to better situate the present findings in the literature, even as the authors already do this well in the Introduction and Discussion. With respect to tiredness and sleep, please allude to current scholarship on peripartum impacts on men's sleep patterns. Given the point that fathers want "frank discussions" the authors might also touch on sexual activity, as other literature points to that as part of renegotiating relationships (one example in the literature that could be cited is: Gray et al. 2015. Sexuality among fathers of newborns in Jamaica. BMC Pregnancy and Childbirth 15: 44). The observations about space for fathers at birth can be placed in broader cultural and historic context; this is a
--

	recent development for fathers to be bed-side in a clinical setting at the time of a child's birth. The authors might cite for this point Huber & Breedlove 2007 Cross-Cultural Research 41: 196-219 for empirical work on men's roles in birth cross-culturally, as also cited in Gray and Anderson. 2010. Fatherhood: Evolution and Human Paternal Behavior book. To the take-away point on p. 30 about what fathers say they want, can other research be cited that addresses what sorts of outreach or information has differential uptake among men during antenatal and postpartum phases? Are men more reachable if targeted alone vs. as a unit with a partner and child? While interviews indicated some of the fathers these men felt could impact their use of resources, the distinction between what men say and do plus efficacy of variable outreach attempts would help address the likelihood of use of newly deployed resources.
--	--

REVIEWER	Gary Clapton University of Edinburgh, Scotland
REVIEW RETURNED	07-Jun-2019

GENERAL COMMENTS	This is a fine piece of work with a well-announced focus, methodology and analysis. Its scope spans ground that has been covered in a variety of other publications and research studies, however, it provides a highly useful and up-to-date account of the persistence of problems previously identified, especially the solely mother-orientated nature of maternity services eg. 'Who's the Bloke in the Room?' by Burgess and Goldman 2018. In the 'barriers' section, more might have been made of the feminisation of reception areas, foyers, waiting rooms and the lack of images of fathers in the publicity for the various services referred to). This is a minor quibble about a readable and stimulating work.
---

REVIEWER	Susan J Rees University of New South Wales
REVIEW RETURNED	16-Jul-2019

GENERAL COMMENTS	The study is important to the field and I congratulate the authors. The findings are not well aligned with the aim, which is ill-defined and doesn't adequately guide the study. The result is that the paper is too comprehensive and lacks a consistent and defined narrative, as well as clear and persuasive outcomes that resonate with the objectives. The study and presentation of findings, according to the author's literature review, could have focused more on describing the 'needs' - what fathers want, how should it be provided, who should provide it, when is optimal? As it is the findings have generated some important commentary on how men think about being fathers, their experiences of fatherhood, and what their needs are. These are, however, three different areas for inquiry and if they all remain in this paper they need to be identified and described systematically in the context of answering well defined research question/s. The authors also need to say something about why they emphasised cultural diversity and yet didn't mention what if any difference that made to the findings (they mention English language didn't make a substantive difference but I note almost all had English as a first language). The paper also needs editing and some attention to expression.
--

VERSION 1 – AUTHOR RESPONSE

No.	Feedback	Actions/ Comments
1.	Reviewer 1: With respect to limitations and methods, this is obviously a non-representative sample, which raises questions about generalizability, but the authors already acknowledge that. Still, it appeared that only one father was younger than 30 years of age; in the Discussion, the authors might note that some literature highlights paternal age (e.g., teenage fathers vs. older fathers in say 50s+) as a salient variable to paternal experience itself.	Additional information has been included in the limitations about only having 1 father under 30 years of age and why this may be the case for this study (pg- 38)
2.	More information about participation rates (e.g., number of invited participants, refusals, etc.) could be provided.	It is not possible to provide details of the number of participants invited/ refused due to the recruitment approach used. All fathers meeting study inclusion criteria between Sept 2017 and Feb 2018 were invited by local health visitors across selected study sites. In addition, posters were displayed at various health and children’s centres so men could volunteer to participate. Recruitment approaches have been described in detail in the ‘Methods’ section under ‘Participants’. Additional information has been added on how many men who had information about the study contacted the researcher, and how many met study inclusion criteria and were recruited (pg - 8)
3.	The time-frame (first-time fathers with a child less than 12 months of age) could encompass quite variable post-partum experiences, durations since birth, etc., which is another limitation.	This has been acknowledged further in study limitations, pg - 38
4.	In the Discussion, a few more contextual references could be provided to better situate the present findings in the literature, even as the authors already do this well in the Introduction and Discussion. With respect to tiredness and sleep, please allude to current scholarship on peripartum impacts on men's sleep patterns. Given the point that fathers want "frank discussions" the authors might also touch on sexual activity, as other literature points to that as part of renegotiating relationships (one example in the literature that could be cited is: Gray et al. 2015. Sexuality among fathers of newborns in Jamaica. BMC Pregnancy and Childbirth 15: 44).	We have not included these references as the ‘frank discussions’ fathers talked about mainly related to better preparation about the challenges of early fatherhood. We considered the paper on sexuality among fathers in Jamaica and felt that the findings may not necessarily apply to UK fathers.

5.	The observations about space for fathers at birth can be placed in broader cultural and historic context; this is a recent development for fathers to be bed-side in a clinical setting at the time of a child's birth. The authors might cite for this point Huber & Breedlove 2007 Cross-Cultural Research 41: 196-219 for empirical work on men's roles in birth cross-culturally, as also cited in Gray and Anderson. 2010. Fatherhood: Evolution and Human Paternal Behavior book.	We have not included these references in the study. Cross-cultural issues were not a focus of father's discussions in this study and therefore we felt they were not relevant to include here. Furthermore, the paper is already long, and including such discussions would increase the length of the paper significantly, while not necessarily adding to relevant discussions relating to the study findings. As per the editor's comments, we have only considered the addition of references suggested by the reviewers if they were relevant to the study.
6.	To the take-away point on p. 30 about what fathers say they want, can other research be cited that addresses what sorts of outreach or information has differential uptake among men during antenatal and postpartum phases? Are men more reachable if targeted alone vs. as a unit with a partner and child? While interviews indicated some of the fathers these men felt could impact their use of resources, the distinction between what men say and do plus efficacy of variable outreach attempts would help address the likelihood of use of newly deployed resources.	Revisions have been made, based on this comment but the authors felt this would fit better in the 'Discussion' section rather than the 'Results' section. Please see revisions on pg- 35-36
7.	Reviewer: 2 In the 'barriers' section, more might have been made of the femimisation of reception areas, foyers, waiting rooms and the lack of images of fathers in the publicity for the various services referred to). This is a minor quibble about a readable and stimulating work.	Fathers in this study did not specifically mention feminisation of reception areas, foyers, waiting rooms and the lack of images of fathers etc. Therefore, this has not been discussed in the paper. Men however talked about the lack of father-friendly facilities on labour and postnatal wards which has been discussed and highlighted in the 'Discussion' section (pg-32)

8.	Reviewer: 3 The findings are not well aligned with the aim, which is ill-defined and doesn't adequately guide the study. The result is that the paper is too comprehensive and lacks a consistent and defined narrative, as well as clear and persuasive outcomes that resonate with the objectives.	Thank you for these comments. While we have considered your comments, in our view findings do answer the research questions and are clearly aligned to the aims. Given this, and the very positive feedback from the other reviewers (see below), we do not consider that we need to revise our paper in line with this feedback. Reviewer 1: "This is an excellent qualitative study of a sample of 21 first-time fathers in the UK. The methods are described well and appropriate to the study aims. The results are organized into 9 key themes, with illustrative quotes taken from interviews provided for these themes. The work is situated well within the literature, both in the Introduction and Discussion sections." Reviewer 2: "This is a fine piece of work with a well-announced focus, methodology and analysis. Its scope spans ground that has been covered in a variety of other publications and research studies, however, it provides a highly useful and up-to-date account of the persistence of problems previously identified, especially the solely mother-orientated nature of maternity services".
9.	The study and presentation of findings, according to the author's literature review, could have focused more on describing the 'needs' - what fathers want, how should it be provided, who should provide it, when is optimal? As it is the findings have generated some important commentary on how men think about being fathers, their experiences of fatherhood, and what their needs are. These are, however, three different areas for inquiry and if they all remain in this paper they need to be identified and described systematically in the context of answering well defined research question/s.	While one of the aims of the study was to focus on the gaps identified in the literature review (i.e. what fathers want, how should it be provided, who should provide it, when is optimal), the overall aim was to identify how men experienced first-time fatherhood as well as what their perceived needs were during this period. Therefore, the paper includes comprehensive findings relating to their experiences during the perinatal period. 'Men's perceived needs: what fathers want' was one of the categories that has been described in detail.
10.	The authors also need to say something about why they emphasised cultural diversity and yet didn't mention what if any difference that made to the findings (they mention English language didn't make a substantive difference but I note almost all had English as a first language).	Of the 21 men interviewed, four (nearly 20%) did not have English as their first language. While the men came from various cultural backgrounds, there were no differences noted in the findings. This was clearly stated at the end of the discussion section (pg-38), as follows: There were no substantive patterns identified between the different groups of fathers. For example, there were no obvious linkages noted for ethnicity, age, income or education of participants. Resident and non-resident fathers interviewed highlighted broadly similar needs, as did fathers for whom English was a first language and those for

		whom it was not. As this study only included first time biological fathers, the mental health and wellbeing needs and experiences of subsequent fathers, and non-biological fathers remain unknown, areas which need to be addressed in future research.
11.	The paper also needs editing and some attention to expression.	We have re-read this paper several times and remain unclear about where editing or amendments are necessary. More clarity around this from the journal Editors would be appreciated.

VERSION 2 – REVIEW

REVIEWER	Susa Rees UNSW Sydney Australia
REVIEW RETURNED	10-Aug-2019

GENERAL COMMENTS	I have made another comment about perceived representativeness, something that should be made clear was not an objective. It needs to be stated that the study size and method didn't allow for examination for associations related to demographic differences. The discussion should link back more systematically to the themes and data, which is rich and could be examined more fully to answer the research questions and to consider and discuss in that context. The study nevertheless provides some insight into some issues for male parents. The reviewer provided a marked copy with additional comments. Please contact the publisher for full details.
--

VERSION 2 – AUTHOR RESPONSE

Reviewer Feedback	Actions/ Comments (Authors)
I have made another comment about perceived representativeness, something that should be made clear was not an objective. It needs to be stated that the study size and method didn't allow for examination for associations related to demographic differences.	Thank you for this comment. This has been incorporated on page 3, as suggested by the reviewer.
The discussion should link back more systematically to the themes and data, which is rich and could be examined more fully to answer the research questions and to consider and discuss in that context. The study nevertheless provides some insight into some issues for male parents.	We have carefully considered the reviewer's comments and revisited our discussion in light of this feedback. We remain of the view that the discussion does link systematically to the themes and data – throughout the discussion, we refer to our findings and explain these in terms of our research questions and the context of current maternity care. As such, we do not feel that this needs to be further strengthened and hope that this is acceptable to the Editors.